# Precise cooperative sulfur placement leads to semi-crystallinity and selective depolymerisability in CS₂/oxetane copolymers

Christoph Fornacon-Wood [1], Bhargav R. Manjunatha [1], Merlin R. Stühler[1], Cesare Gallizioli[1], Carsten Müller[1], Patrick Pröhm[1] & Alex J. Plajer [1] ✉

CS₂ promises easy access to degradable sulfur-rich polymers and insights into how main-group derivatisation affects polymer formation and properties, though its ring-opening copolymerisation is plagued by low linkage selectivity and small-molecule by-products. We demonstrate that a cooperative Cr(III)/K catalyst selectively delivers poly(dithiocarbonates) from CS₂ and oxetanes while state-of-the-art strategies produce linkage scrambled polymers and heterocyclic by-products. The formal introduction of sulfur centres into the parent polycarbonates results in a net shift of the polymerisation equilibrium towards, and therefore facilitating, depolymerisation. During copolymerisation however, the catalyst enables near quantitative generation of the metastable polymers in high sequence selectivity by limiting the lifetime of alkoxide intermediates. Furthermore, linkage selectivity is key to obtain semi-crystalline materials that can be moulded into self-standing objects as well as to enable chemoselective depolymerisation into cyclic dithiocarbonates which can themselves serve as monomers in ring-opening polymerisation. Our report demonstrates the potential of cooperative catalysis to produce previously inaccessible main-group rich materials with beneficial chemical and physical properties.

The incorporation of sulfur centres into the polymer main chain rationally offers distinct material properties and functions associated with the periodic trends of the main-group elements[1]. The altered electronic nature of sulfur containing polymers compared to their lighter oxygen analogues causes amongst others degradability benefits, the potential to scavenge heavy metal contaminants, high refractive indices, stimuli responsiveness in drug delivery systems and enhanced semi-crystallinity[2–7]. Moving from oxygenated to sulfurated polymers can furthermore be used to tune the (de)polymerisation equilibria of polymers to facilitate chemical polymer recycling[8–10]. Furthermore, using monomers which can be directly sourced from elemental sulfur (such as CS₂ from S₈ and methane) is also relevant in the context of S₈ utilisation, a waste product of the petrochemical industry being produced at an annual megaton surplus[11–15]. Sulfur containing polymers are often synthesised by condensation or ring-

opening polymerisation (ROP) methods, which can require multi-step monomer synthesis and do not give easy access to many polymer structures[16,17]. An increasingly popular method for the synthesis of heteroatom containing polymers is the ring-opening copolymerisation (ROCOP) of a strained heterocycle with heteroallenes. Having gained prominence for the selective copolymerisation of CO₂ with epoxides or oxetanes to polycarbonates, ROCOP cannot only make use of under-utilised monomer feedstocks but also produce previously inaccessible polymer structures[18–22]. Taking this methodology to the area of sulfurated polymers, ROCOP of carbonyl sulfide COS with epoxides or oxetane represents a well-established route to selectively yield poly(monothiocarbonates) (Fig. 1(a))[23–29]. Yet this ROCOP requires specialised steel reactor set-ups to safely handle highly toxic, flammable and gaseous carbonyl sulfide as well as commercial access to high purity COS (which is for example currently not the case in Europe) limiting

[1]Intitut für Chemie und Biochemie., Freie Universität Berlin, Fabeckstraße 34-36, 14195 Berlin, Germany. ✉e-mail: plajer@zedat.fu-berlin.de

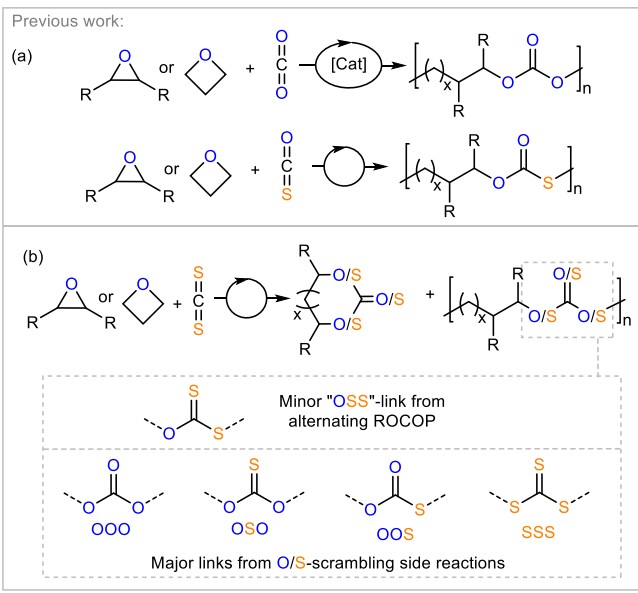

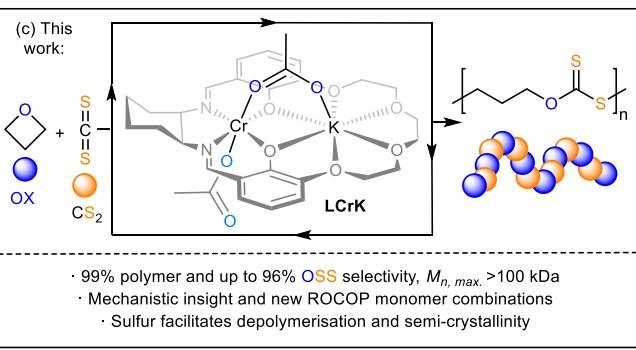

**Fig. 1 | Outline of the presented work.** Comparison of products formed during (**a**) (CO₂ or COS)/(epoxide or oxetane) and (**b**) CS₂/(epoxide or oxetane) ROCOP. **c** Selective CS₂/oxetane ROCOP yielding poly(dithiocarbonates) presented in this report. x = 0 (R = H), 1 (R = H, alkyl).

the utility of this ROCOP in standard research laboratories. Using the liquid heavier homologue CS₂ is substantially simpler as it can be employed as a cosolvent and in principle, allows access to copolymers with even higher sulfur ranks. However, the ROCOP of CS₂ with oxygen containing heterocycles has so far been plagued with numerous side reactions as shown in Fig. 1(b). Polymers comprising different mono- (-O-C(=S)-O- OSO, O-C(=O)-S- OOS) di- (-O-C(=S)-S- OSS) and trithio-carbonate (-S-C(=S)-S- SSS) as well as all-oxygen carbonate (-O-C(=O)-O- OOO) and thioether links are always obtained in addition to small-molecule by-products, such as different cyclic (thio)carbonates, thiir-anes and COS instead of the expected poly(dithiocarbonates) from alternating ROCOP as demonstrated, assigned and rationalised by the groups of Wan, Zhang and Darensbourg[30–38]. Polymer selectivities versus cyclic heterocarbonates range from 30 to 80% in which dithiocarbonate links from alternating copolymerisation make up the minor (typically 0–30%) fraction. Product distribution and polymer composition strongly depend on the exact reaction conditions making material properties unpredictable and necessitating time-consuming purification steps. The process behind the formation of the different linkages has been termed O/S-scrambling and is suspected to originate from the attack of metal-alkoxide chain-ends (formed from epoxide/oxetane ring-opening alike intermediate A in Fig. 2) into dithiocarbo-nate linkages in place of propagation with CS₂ (forming metal-dithiocarbonate chain ends alike DTC in Fig. 2). Notably this side-process extends to related polymerisations and causes linkage scrambling and small-molecule by-products[39–46]. Previously it has been

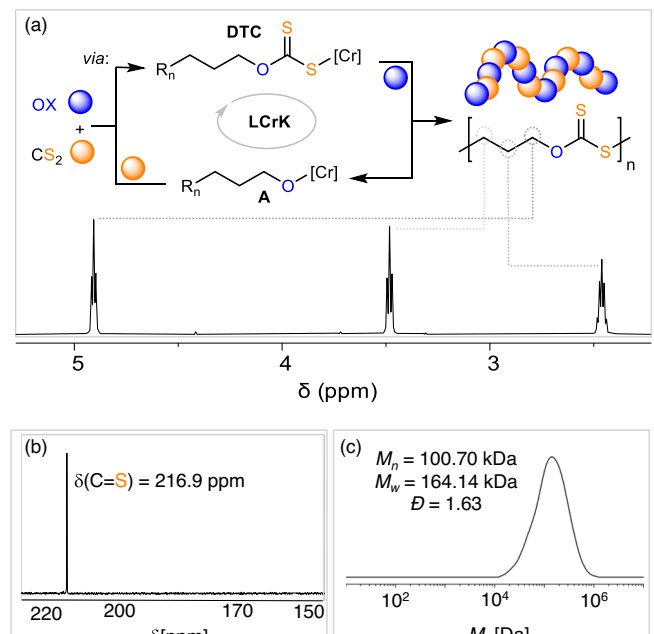

**Fig. 2 | Polymer characterisation. a** CS₂/OX ROCOP and ¹H (CDCl₃, 400 MHz), (**b**) ¹³C (CDCl₃, 126 MHz) NMR as well as (**c**) GPC trace of the isolated polymer corresponding to Table 1 run #1 (**a**, **b**) and #10 (**c**). $R_n$ denotes growing polymer chain.

shown for all-oxygen ROCOP that cooperative catalysts can control the reactivity of alkoxide intermediates and limit or even productively regulate side reactions[47–54]. Williams and co-workers in particular showed that a heterobimetallic Co(III)K catalyst can limit back-biting side reactions originating form alkoxides in CO₂/epoxide ROCOP which lead to excellent polymer and linkage selectivities[55]. Relatedly, we found in a comparative study on CO₂/epoxide versus CS₂/epoxide ROCOP employing a series of heterobimetallic Cr(III)-alkalimetal complexes based on a bis-methoxy substituted SalcyCr(III) complex that cooperativity is maintained moving from CO₂ to CS₂ ROCOP[56]. Yet these catalysts still yielded product mixtures in which the OSS link from alternating ROCOP represented the minor product calling for further catalyst development. In terms of targeted polymer structures, we hypothesised that linkage control is particularly important for unsubstituted monomers (such as ethylene oxide or oxetane) as the material properties of the resulting polymers are predominantly determined by the chemical nature of the linkages. In the case of oxetane (OX) previous reports by Darensbourg and Zhang showed that sulfur enhanced semi-crystallinity is observed in the COS/OX copoly-mers while the same was not the case for the CS₂/OX copolymers[28,34]. We inferred this to stem from the uncontrolled microstructure of this polymer as a consequence of the O/S scrambling process. Hence this monomer combination represents an opportunity for selective cata-lysis to potentially control material properties which we report in this contribution.

## Results

### CS₂/OX ROCOP by LCrK

Motivated by the precedence for chromium(III)-salen catalysts in CS₂/oxetane (CS₂/OX) ROCOP and work on potassium based systems within heterobinucleating ligands alike **L**, a heterobimetallic Cr(III)/K complex **LCrK** was prepared by one-pot template condensation of **LH₂** with diamino-cyclohexane, KOAc and Cr(OAc)₂ following aerobic oxi-dation (see Supplementary Notes 2 in the Supplementary Information)[34,55,57]. The complex was obtained as the water adduct **LCrK·H₂O** and characterised by elemental analysis, IR spectroscopy and HR-ESI mass spectrometry. Attempts to remove H₂O via

**Table 1 | CS$_2$/OX ROCOP with LCrK**

| Run | Cat:OX:CS$_2$ | T [°] | t [h] | TON[a] | Polymer [%] | OSS [%][b] | $M_n$ [kDa] (Đ)[c] | $M_{n,theo}$[d] |
|---|---|---|---|---|---|---|---|---|
| #1 | 1:1000: 2000 | 80 | 2 | 820 | 99 | 96 | 14.03 (1.32) | 55.09 |
| #2 | 1:1000: 1000 | 80 | 1.5 | 930 | 99 | 86 | 17.04 (2.23) | 62.47 |
| #3 | 1:1000: 4000 | 80 | 2 | 950 | 99 | 95 | 15.66 (1.75) | 63.81 |
| #4 | 1:1000: 2000 | 40 | 24 | 400 | 99 | 96 | 5.20 (1.37) | 26.90 |
| #5 | 1:1000: 2000 | 60 | 20 | 1000 | 99 | 95 | 14.23 (1.84) | 67.17 |
| #6 | 1:1000: 2000 | 90 | 0.75 | 750 | 99 | 94 | 11.62 (1.40) | 50.39 |
| #7 | 1:1000: 2000 | 110 | 0.25 | 980 | 99 | 88 | 13.26 (1.75) | 65.82 |
| #8 | 1:1000: 2000 | 80 | 1.5 | 1000 | 99 | 96 | 29.20 (1.83) | 67.17 |
| #9 | 1:4000: 8000 | 80 | 2 | 3800 | 99 | 96 | 79.88 (1.67) | 255.06 |
| #10 | 1:10$^4$: 2×10$^4$ | 80 | 4 | 7300 | 99 | 96 | 100.70 (1.63) | 489.93 |
| #Lit.[34]* | 1:1000: 2000 | 80 | 12 | 1000 | 64 | 6 | 6.50 (1.76) | 43.00 |

[a]Turnover number (TON), number of equivalents of OX consumed per equivalent of catalyst.
[b]Relative integrals, in the normalised $^1$H NMR spectrum of CH$_2$ resonances due OSS versus other heterocarbonate links.
[c]Determined by GPC (gel permeation chromatography) measurements conducted in THF, using narrow MW polystyrene standards to calibrate the instrument.
[d]Calculated assuming initiation of both acetate coligands. OX purified over CaH$_2$ for run #1-#7 and successively over CaH$_2$ and Na for run #8-#10.

prolonged drying under dynamic vacuum at elevated temperatures were unsuccessful. Testing **LCrK** at previously optimised reaction conditions (1 eq. catalyst: 1000 eq OX: 2000 eq. CS$_2$ and 80 °C, Table 1 run#1, Fig. 2, Supplementary Notes 3 in the Supplementary Information) in a melamin capped vial with a teflon inlay yielded a highly viscous product mixture in which stirring stopped after 2 h[34,35,39]. Cooling to RT caused solidification via spherulite formation as seen by optical microscopy. Analysis of the product mixture by $^1$H NMR spectroscopy reveals 99% formation of poly(heterocarbonate), only trace amounts of small-molecule by-products and no evidence of homopolymer links. The polymer can be easily isolated by centrifugation and air drying and is obtained as a pale-yellow powder ($M_n$ = 14.03 kDa, Đ = 1.32). It comprises ca. 96% OSS links ($\delta$(C$^q$=S) = 216.9 ppm, $\tilde{v}$(C=S) = 1036.1 cm$^{-1}$) and 4% scrambled links as quantified by $^1$H NMR spectroscopy. Such high selectivity represents a significant improvement over state-of-the-art catalysis which produces cyclic heterocarbonate by-products (36%) and polymers only comprising 6% OSS links under the same conditions (Table 1 run #Lit.). To explore our new methodology further, we next assessed the effects of different reaction conditions and again observed 99% polymer formation in all cases. As seen in Table 1 lower CS$_2$ loadings lead to more O/S-scrambling while higher CS$_2$ loadings do not improve the OSS selectivity (run #2, #3). Therefore, we continued our study with a 1:2 OX:CS$_2$ ratio to maximise selectivity but avoid excessive dilution (vide infra). Rates increase with temperature (maximum TOF of 3920 h$^{-1}$ at 110 °C), while OSS selectivity decreases with increasing temperature indicating that O/S scrambling is a thermodynamically favourable process. Attempts to obtain MALDI or ESI-MS data to investigate the polymer end-groups were unsuccessful. However, IR spectroscopy shows some ester end-groups ($\tilde{v}$(C=O) = 1720.0 cm$^{-1}$) from acetate initiation to be part of the CS$_2$/OX copolymer which could be unambiguously identified by $^1$H-$^{13}$C HMBC ($\delta$(C$^q$=O, Ester) = 171.0 ppm correlating to the OAc methyl group). Furthermore, the $^{31}$P-NMR end-group test (Supplementary Figs. 20 and 21 in the Supplementary Information) allowed us to clearly identify primary alcohol chain ends and this suggests initiation via OX ring opening by the acetate coligands of **LCrK** and termination by protonation during work-up forming α-OAc,ω-OH-functional chains. Yet the $^{31}$P-NMR end-group test also indicates the formation of some α,ω-OH-telechelic chains via initiation from H$_2$O or diol impurities. Accordingly, although molecular weights correlate to some extent with TON (i.e. lower TON in run #4 resulting in lower $M_n$) and $M_n$ increases with reaction progress, theoretical (assuming initiation of both acetates) and observed molecular weights deviate significantly. We suspected that this is presumably due to chain transfer events with

e.g. catalyst bound water or diol impurities in OX which is commonly observed in ROCOP[18]. We hypothesised that the latter might occur, even though OX for run #1-#7 had been purified over CaH$_2$. Hence, we employed OX which had been successively purified over CaH$_2$ and elemental Na (run #8) in a repeat to run #1. We indeed observed an approximate doubling of the obtained molecular weight ($M_n$ = 14.03 kDa in run #1 to $M_n$ = 29.20 kDa in run #8) which, although improved, is still lower than $M_{n,theo}$ (67.17 kDa) even when taking chain-transfer with the catalyst bound H$_2$O into account ($M_{n,theo}$ = 44.80 kDa) and this deviation becomes more pronounced for lower catalyst loadings. Other attempts such as drying over molecular sieves or testing OX from different suppliers to remove and avoid water or other protic impurities did not result in further improvement letting us hypothesise that side reactions other than chain transfer (such as cyclic polymer formation; vide infra) may also limit the molecular weights[18,46,58]. **LCrK** also shows excellent performance (max. TON 7300) and activity at loadings as low as 0.003 mol% versus liquid monomers (run #10) producing polymers with a high maximum $M_n$ = 100.70 kDa (Đ = 1.63). State-of-the-art catalysis previously only achieved a maximum $M_n$ = 13.7 kDa (Đ = 1.70)[34].

## Structure-selectivity and activity studies

Next, we wondered whether the excellent performance of **LCrK** is due to favourable electronic matching of the employed metals evoking cooperativity or due to the circumstance that a single-component system is employed. Shedding light on this question (Supplementary Notes 4 in the Supplementary Information, Table 2, Fig. 3) we prepared the Na (**LCrNa**) and Rb (**LCrRb**) derivative and employed them in CS$_2$/ OX ROCOP (Table 2, run #1 and #2). While LCrRb performs similar to **LCrK**, the sodium derivative **LCrNa** produces more scrambled (30%, primarily SSS) links and also substantial amounts (ca. 20%) of cyclic carbonate by-products. Selection of the transition metal is equally vital as moving to **LZnK** (run #3) featuring Zn(II), a popular metal in ROCOP catalysis, while keeping the active K results in no activity[52,59–61]. To mimic the electronic and coordinative situation in **LCrK** albeit as distinct components, we then employed a bicomponent catalyst comprising a bis-methoxy substituted (MeO)$_2$SalCyCrOAc **L'Cr** complex (Fig. 3) with KOAc@18-crown-6. This led to a significant increase of scrambled links from 4 to 47% (Table 2, run #4) alongside 20% cyclic carbonate by-products. While KOAc@18-crown-6 by itself is completely inactive (run #5), **L'Cr** by itself produces even more scrambled polymers (8% OSS links, run #6). Using **L'Cr** with PPNOAc, in which the cocatalytic salt features the weakly coordinating bis(triphenylphosphine)iminium cation, results in 90% scrambling and 20% cyclic

## Table 2 | CS2/OX ROCOP with alternative catalysts

| Run | [Cat] | t [h][c] | TON[a] | Polymer [%] | OSS [%][b] | $M_n$,[kDa] $(Đ)^c$ |
|---|---|---|---|---|---|---|
| #0 | **LCrK** | 2 | 820 | 99 | 96 | 14.03 (1.32) |
| #1 | **LCrRb** | 1.5 | 980 | 99 | 94 | 15.82 (1.62) |
| #2 | **LCrNa** | 17 | 1000 | 80 | 70 | 12.16 (1.91) |
| #3 | **LZnK** | 24 | – | – | – | – |
| #4 | **L'Cr** + KOAc@18c6 | 24 | 1000 | 80 | 47 | 12.41 (1.70) |
| #5 | KOAc@18c6 | 24 | – | – | – | – |
| #6 | **L'Cr** | 24 | 470 | 99 | 8 | 8.32 (1.81) |
| #7 | **L'Cr** + PPNOAc | 24 | 950 | 80 | 8 | 11.66 (1.78) |

$T$ = 80 °C. Note that for run #0, #1 and #6 99% polymer selectivity is observed, while for run #2, #4 and #7 only ca. 80% polymer selectivity is observed.

[a]Turnover number (TON), number of equivalents of OX consumed per equivalent of catalyst.
[b]Relative integrals, in the normalised $^1$H NMR spectrum of $CH_2$ resonances due OSS versus other heterocarbonate links.
[c]Determined by GPC (gel permeation chromatography) measurements conducted in THF, using narrow MW polystyrene standards to calibrate the instrument. $M_{n,theo.}$ = 31.54–67.17 kDa.

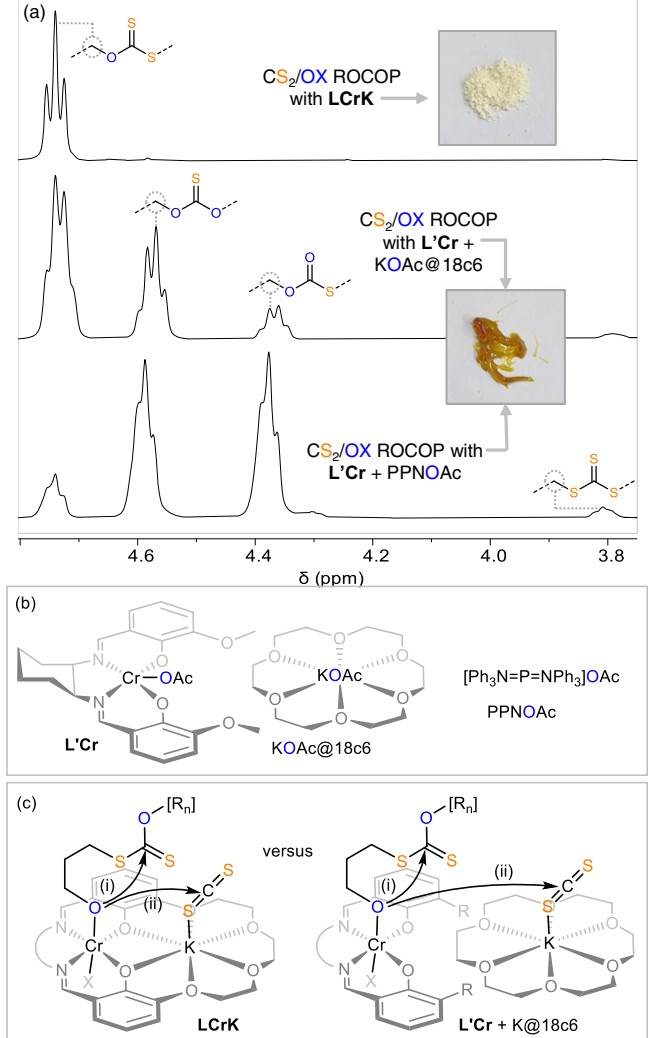

**Fig. 3 | O/S scrambling. a** Zoom into the $^1$H NMR (CDCl$_3$, 400 MHz) spectra of crude mixtures produced by the monocomponent **LCrK** catalyst (Table 1 run #1) and bicomponent variants (Table 2 run #4 and #7) alongside photographs of the produced polymers. **b** Structures of other catalysts. **c** Reaction pathways of propagation versus O/S scrambling for mono and bicomponent catalysts, X = OAc, [R$_n$] = polymer chain.

carbonate byproducts (run #7) suggesting that weakly coordinated chain-ends are responsible for low linkage selectivities. Interestingly, no crystallisation from solution is observed for these scrambled polymers which are obtained as amorphous semi-solids (vide infra for further discussion). It has been previously suggested that side reactions originating from alkoxide intermediates are responsible for the scrambling process[33–35]. Hence, we reacted preformed poly(dithiocarbonate) (100 eq. repeat unit, Table 1 run #1) with **LCrK** (1 eq.) and OX (1000 eq.), conditions under which due to the absence of CS$_2$ only alkoxide intermediates can exist, and indeed observed scrambling of the polymer (from 6 to 90% after 24 h at 80 °C). Notably this also results in a drastic decrease of the molecular weights down to oligomers smaller than 1 kDa suggesting that alkoxide originated polymer attack decreases molecular weights. Importantly this observation could help explain why obtained molecular weights deviate from theoretical ones. As shown in Supplementary Fig. 104 in the Supplementary Information alkoxide addition and elimination to heterocarbonates of the same chain can in principle lead to cyclic polymer formation to decrease molecular weights in addition to chain-transfer events. Combined, our results outlined in this paragraph indicate that **LCrK** suppresses side reactions originating from decoordinated alkoxide chain ends via metal-metal cooperativity. Matching of the correct metals as well as ensuring cooperativity by fixing both active metals within the same ligand scaffold are vital to achieve high sequence selectivity in CS$_2$/OX ROCOP. As depicted in Fig. 3(c) we infer in reference to recent computational studies by Williams et al. on CO$_2$/epoxide ROCOP that alkoxide based reactions originate from chromate centres while heteroallene activation occurs at the alkali metal[57]. Considering that reaction step (i) represents the entry point to O/S scrambling pathways (Supplementary Notes 9 in the Supplementary Information) in reference to Darensbourg and Werner it becomes clear that if both metals are not fixed within the same scaffold the intermolecular propagation pathway (i) has to compete with the intramolecular scrambling pathway (ii) and the same is conceptually the case for other bicomponent catalysts[33–35]. Heterobimetallic **LCrK** is hence inferred to lead to intermolecular propagation (i) and thereby avoid O/S scrambling.

### Monomer scope of LCrK

Next, we investigated the monomer scope of our new methodology (Supplementary Notes 5–7 in the Supplementary Information) and found that **LCrK** is also capable of copolymerising a variety of previously unexplored monomer combination in quantitative polymer selectivity. The 3,3'-disubstituted oxetanes (Table 3, run #1 - #4, Fig. 4) 3,3'-dimethyloxetane (OX$^{Me}$) as well as the ethyl (OX$^{OEt}$), benzyl (OX$^{OBn}$) and allylether (OX$^{OAll}$) substituted methyloxetane undergo CS$_2$ ROCOP yielding polymers with alkylic, arylic and functional olefine substituents. OSS selectivities are somewhat decreased compared to the parent CS$_2$/OX ROCOP (83–91% versus 96%) and in all cases errors comprise in equal parts of SSS linkages and OSO linkages. This can be rationalised by chain-end O/S exchange processes producing OSO-*alt*-SSS links as proposed by Werner and co-workers for CS$_2$/epoxide ROCOP (Supplementary Fig. 104 in the Supplementary Information)[35]. Note that this scrambling mode is distinct from the random scrambling observed in Table 2, for which random rather than equal ratios of the OSO and SSS are observed. Testing **LCrK** in the ROCOP of CS$_2$ with cyclohexene oxide (CHO) also reveals field leading performance (Table 3, run #5). 83% OSS links alongside a random distribution of combined 17% SSS, OSO, OOS and OOO links are observed. In contrast to CS$_2$/OX ROCOP, CS$_2$/CHO ROCP produces ca. 20% cylic dithiocarbonate byproducts. Nevertheless, **LCrK** also outperforms previously reported CS$_2$/CHO catalysts which only achieves ca. 10% OSS linkages in 40–70% polymer selectivity[32,35]. Moving to the technologically relevant epoxide propylene oxide (PO) shows significantly more cyclic carbonate by-

**Table 3 | Monomer scope of LCrK**

| Run | Loading[a] | t [h] | T [°] | TON[b] | Alt. [%][c] | Polymer [%] | $M_n$ [kDa] (Đ)[d] | $T_{m/g/d,5\%}$ |
|---|---|---|---|---|---|---|---|---|
| #0 | 1000 OX: 2000 CS₂ | 2 | 80 | 820 | 96 | 99 | 14.03 (1.32) | $T_m = 89.3\ °C$; $T_g = -16.7\ °C$; $T_d = 163.2\ °C$ |
| #1 | 1000 OX^Me: 2000 CS₂ | 24 | 80 | 1000 | 91 | 99 | 16.81 (1.53) | $T_m = 107.2\ °C$; $T_g = 5.5\ °C$; $T_d = 195.0\ °C$ |
| #2 | 1000 OX^OEt: 2000 CS₂ | 30 | 80 | 1000 | 89 | 99 | 17.88 (1.43) | $T_g = -9.5\ °C$; $T_d = 190.5\ °C$ |
| #3 | 1000 OX^OBn: 2000 CS₂ | 20 | 80 | 850 | 83 | 99 | 20.95 (1.54) | $T_g = 8.0\ °C$; $T_d = 208.0\ °C$ |
| #4 | 1000 OX^OAll: 2000 CS₂ | 1 | 80 | 1000 | 89 | 99 | 24.19 (1.58) | $T_g = -17.0\ °C$; $T_d = 220.0\ °C$ |
| #5 | 1000 CHO: 2000 CS₂ | 8 | 80 | 600 | 83 | 80 | 15.94 (1.71) | $T_g = 87.8\ °C$ $T_d = 148.3\ °C$ |
| #6 | 1000 PO: 2000 CS₂ | 2 | 80 | 1000 | 20 | 30 | n.d. | n.d. |
| #7 | 1000 PO: 2000 CS₂ | 16 | 50 | 480 | 54 | 57 | 6.56 (1.53) | $T_g = 11.5\ °C$ $T_d = 147.3\ °C$ |

$T = 80\ °C$.
[a]Relative to 1 eq. LCrK.
[b]Turnover number (TON), number of moles of OX consumed per mole of catalyst.
[c]"Alternation selectivity": links resulting from formal alternation of the heteroallene/PTA with OX determined by comparing the relative integrals in the normalised ¹H NMR spectrum of CH₂ resonances adjacent to heteroatoms.
[d]Determined by GPC (gel permeation chromatography) measurements conducted in THF, using narrow MW polystyrene standards to calibrate the instrument. Mn,theo. = 32.27–114.07 kDa.

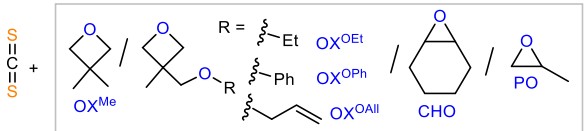

**Fig. 4 | Monomer scope.** Monomer combinations investigated in CS₂ ROCOP with **LCrK** corresponding to Table 3.

products (30% polymer selectivity) and scrambling (20% OSS links) at 80 °C (Table 3, run #6) although decreasing the reaction temperature to 50 °C leads to some improvements to 57% polymer and 54% OSS selectivity (Table 3, run #7). This OSS selectivity exceeds prior literature reports only reaching ≤16% dithiocarbonate links[31,35]. Interestingly, 2D-NMR analysis reveals regioselective PO ring opening at the CH₂ position so that tertiary CHMe carbons sit adjacent to O-atoms in the OSO errors while secondary CH₂ carbons sit adjacent to S-atoms in the SSS errors confirming a scrambling mechanism as proposed by Werner and co-workers (Supplementary Fig. 104 in the Supplementary Information). Testing our catalyst in CS₂/ethylene oxide ROCOP under comparable conditions to PO unfortunately resulted in visible catalyst degradation.

**Sulfuration and sequence-selectivity improve selective depolymerisation**

The circumstance that we observe 99% polymer selectivity for oxetanes irrespective of the employed reaction conditions (Table 1) or oxetane (Table 3) is surprising considering reports concerning the all-oxygen counterpart, i.e. CO₂/OX ROCOP[62]. Here, cyclic trimethylenecarbonate by-products are formed due to establishing (de)polymerisation equilibria between poly(trimethylenecarbonate) and the cyclic six-membered carbonates[45,62–65]. As this observation is most pronounced in geminally disubstituted derivatives due to the Thorpe-Ingold effect, we compared the relative energies of formation of the carbonate polymer carbonate P and the cyclic carbonate **C** from OX^Me and CO₂ as well as the alternating dithiocarbonate polymer P^T and the cyclic dithiocarbonates **C**^T from OX^Me and CS₂ on the B3LYP/cc-pVDZ level of theory (Fig. 5(a₁/₂)) with periodic boundary conditions for **P** and **P**^T (Supplementary Notes 10 in the Supplementary Information), i.e. considering the energy of formation of one repeating unit in an infinite chain. As immediately apparent, CE₂/OX^Me coupling to either cyclic (ΔE = −61 kJ/mol for E = O and −103 kJ/mol for E = S) or poly(dithio)carbonate (ΔE = −116 kJ/mol for E = O and −148 kJ/mol for E = S) is more exergonic for CS₂ than for CO₂, a likely consequence of the decreased stability of a C=E bond versus two C−E

bonds which are broken and formed during the coupling process when moving from O to S[66]. In both cases the polymers **P** or **P**^T are more stable than the respective cyclic forms **C** or **C**^T reflecting *inter alia* the release of ring-strain energy upon polymerisation. However, polymer stability is more pronounced in the all-oxygen case (Fig. 5(a₁), −55 kJ/mol moving from **C** to **P**) than in the sulfurated case (Fig. 5(a₂), −45 kJ/mol moving from **C**^T to **P**^T) which is surprising given the near quantitative polymer selectivity of our CS₂ ROCOP compared to previous CO₂ ROCOPs (*vide supra*) giving cyclic carbonate by-products due to equilibrium limitations[62]. Comparison of the bonding situation in the respective small molecules with the polymers (i.e. **C** with **P** and **C**^T with **P**^T) shows near identical bonding orders in **C** and **P** as well as **C**^T and **P**^T. This suggests that not electronic changes, but rather different ring strain energies of **C** and **C**^T are responsible for the differences in poly- versus cyclic (dithio) carbonate stability which we infer is due a greater tolerance towards bond-angle variation when moving down the periodic table[67]. Combined, these results imply that the (de)polymerisation equilibrium of the **P**^T⇌**C**^T copolymer must lie more on the cyclic (dithio)carbonate side then for the all-oxygen equilibrium **P**⇌**C** under identical conditions. In this respect, Endo and co-workers reported that catalytic KOᵗBu in THF reconstitutes equilibria states of isolated **P**, i.e. starting from 100% **P** at 0.45 M in THF and 20 °C resulted in 7% **C** and 93% **P** at equilibrium[63]. Subjecting isolated the CS₂/OX^Me copolymer **P**^T* (i.e. **P**^T with 92% OSS and 8% OSO-*alt*-SSS links; Table 3, run #1) to analogous conditions results in >99% depolymerisation into ca. 95% **C**^T and ca. 5% unidentified by-products and this confirms sulfur enhanced depolymerisability as predicted by DFT. Accordingly, our other CS₂/OX^OR polymers (Table 3, run #2 – #4) likewise undergo >99% depolymerisation under the above conditions into cyclic dithiocarbonates, while related polycarbonates show equilibrium limitations[62,63]. Note that depolymerisation results in some reversion of the O/S exchange process and reshuffles the O and S centres from the OSO-*alt*-SSS links into cyclic dithiocarbonates containing one O and two S atoms offering easy synthetic access to this class of heterocycles. The CS₂/OX copolymer likewise undergoes depolymerisation albeit necessitating higher dilution (0.023 M) for >99% depolymerisation. While depolymerisation of the nearly alternating copolymer (Table 2, run #0) forms cyclic dithiocarbonates in >95% selectivity, depolymerisation of the randomly scrambled copolymer (Table 2, #7) forms a complex inseparable mixture of products. This implies that the sequence control achieved by **LCrK** is key to enabling chemoselective depolymerisation of the CS₂/OX copolymers. The KOᵗBu catalysed (de)polymerisation mechanism has previously been proposed to involve alkoxide chain-ends that backbite into carbonates links during depropagation and attack into cyclic carbonate monomers during propagation[63]. We probed this hypothesis by attempting to

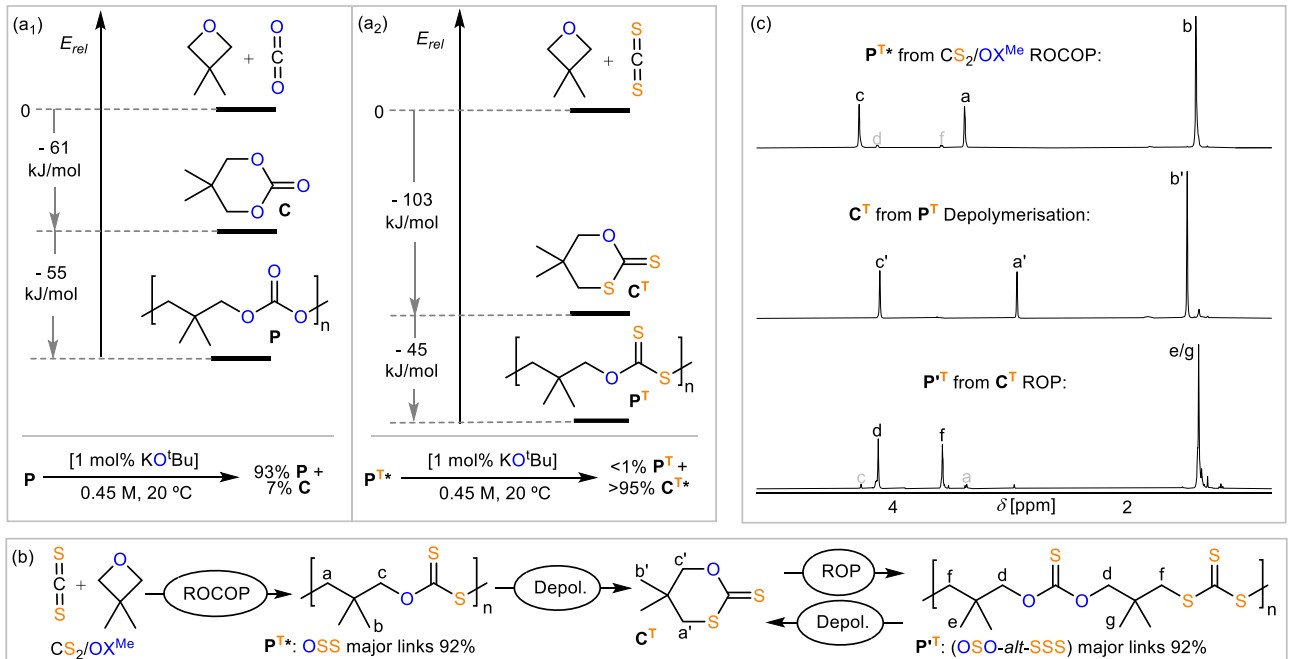

**Fig. 5 | De and Repolymerisation.** Relative energies on the B3LYP/cc-pVDZ level of theory of cyclic versus polymer formation for the (**a₁**) all-oxygen $CO_2$/OX^Me and (**a₂**) the sulfurated $CS_2$/OX^Me cases as well as experimentally determined (de)polymerisation equilibria states under specified conditions; all-oxygen carbonate data displayed in reference to Endo et al.[63]. *note that **P^T** as employed in the depolymerisation errors contains 8% OSO-*alt*-SSS errors; >99% depolymerisation products comprise of ca. 95% **C^T** and 5% unidentified by-products. **b** ROCOP into depolymerisation into ROP sequence. ROCOP conditions as per Table 3 run #1; depolymerisation conditions as per (**a**); ROP conditions 1 eq. 1,5,7-triazabicyclo[4.4.0]dec-5-en (TBD): 1 BnOH: 100 eq. **C^T**, 4 M in DCM, 20 °C, 16 h, 45% conversion, obtained polymer **P^T**: $M_n = 4.50$ kDa, Đ = 1.22, $M_{n,theo} = 7.29$ kDa. **c** Overlaid ¹H NMR spectra (400 MHz, CDCl₃, 25 °C) of **P^T*** produced by $CS_2$/OX^Me ROCOP, **C^T** produced by depolymerisation and **P'^T** produced by **C^T** ROP.

perform **P^T*** depolymerisation in a 1:1 THF:$CS_2$ mixture, in which metal-alkoxide chain-ends should insert $CS_2$ forming less nucleophilic metal-dithiocarbonate chain-ends and indeed did not find any depolymerisation to occur. In accordance with thermodynamically favoured depolymerisation, attempted ring-opening polymerisation of **C^T** at 0.45 M in THF does not occur, conditions under which **C** undergoes polymerisation[63]. Nevertheless repolymerisation of **C^T** can be achieved by solvent choice and concentration. Increasing the concentration to 4 M in DCM with a 1,5,7-triazabicyclo[4.4.0]dec-5-en (TBD) and BnOH catalyst (1 TBD: 1 BnOH: 100 **C^T**, RT, 16 h) sufficiently shifts the (de)polymerisation equilibrium to achieve 45% ROP after 16 h ($M_n = 4.50$ kDa, Đ = 1.22, $M_{n,theo} = 7.29$ kDa) (Fig. 5(b/c)). The ROP polymer **P'^T** comprises OSO-*alt*-SSS (92%) and OSS (8%) links, which can be rationalised by chain-end alkoxide originated O/S exchange processes in reference to related reports by Buchard et al.[45,46]. Similar to **P^T*** ($T_m = 107.2$ °C $T_g = 5.5$ °C), **P'^T** is semi-crystalline ($T_m = 54.7$ °C $T_g = −10.0$ °C) albeit with significantly lowered melting temperature by ca. 50 °C again highlighting how sequence selectivity achieved in the first place by **LCrK** enhances material properties. Subjecting **P'^T** to the depolymerisation conditions outlined above likewise results in >99% depolymerisation under reversion of the O/S exchange process yielding **C^T** highlighting the potential of our copolymers in the context of chemical recycling[68]. Taken together our results indicate that $CS_2$/OX ROCOP generates poly(dithiocarbonates) which are in a metastable or non-equilibrated state as a consequence of a sulfur induced ring-strain decrease. However, in ROCOP with **LCrK** metal-dithiocarbonate formation at the propagating chain-end with $CS_2$ limits the lifetime of metal-alkoxide intermediates which are responsible for establishing (de)polymerisation equilibria into cyclic dithiocarbonate and for O/S scrambling. This does not (efficiently) occur in other ROCOPs and ROPs targeting related poly(thio)carbonates which gives reasons behind the selectivity benefits of **LCrK** over existing methodologies.

## Sulfuration and sequence-selectivity lead to semi-crystallinity
As mentioned above, the high linkage selectivity achieved by **LCrK** can have a strong influence on the thermal properties of the obtained polymers. The $CS_2$/OX copolymers with high OSS selectivity (i.e. Table 1) crystallise from the reaction mixture at room temperature and hence exhibits semi-crystallinity in the solid state as shown by DSC ($T_m = 89.3$ °C, $\Delta H_m = 36.01$ J/g, $T_c = 47.0$ °C for Table 1 run #1 in the 2nd DSC heating cycle, Fig. 6) and PXRD (primary reflections are observed at $\theta = 19.7°$, 22.13° and 27.54°). Neither solution nor solid-crystallisation in DSC measurements is observed for the O/S-scrambled polymers (Table 2 run #4 -#7) which are obtained as amorphous materials after removal of excess monomers and do not exhibit any semi-crystallinity by DSC ($T_g = −29.1$ °C for Table 2 run #7). Owing to the low glass-transition temperature of these material, linkage selectivity is hence key to allow these materials to be hot pressed into free-standing and shape persistent films. Uniaxial tensile testing on films prepared from the polymer corresponding to Table 1 run #9 shows a break at 3.6% elongation and a tensile strength of 20.6 MPa. Furthermore, dynamic mechanical analysis (DMA) shows a thermomechanical spectrum typical of a semi-crystalline material. A high maximum storage modulus of $E' = 4.2$ GPa is observed in the glassy state which decreases to $E' \approx 0.7 - 0.3$ GPa after an alpha glass transition at $T_a = −14.7$ °C (defined as the maximum of $\tan(\delta)$, $\delta = E''/E'$, Fig. 6b) which corresponds well to the $T_g$ observed by DSC. A rubbery plateau is observed until the flow temperature at about 85 °C is reached for the semi-crystalline poly(trimethylenedithiocarbonate). Note that the all-oxygen analogue poly(trimethylenecarbonate) obtained from $CO_2$/OX ROCOP has also been reported to be an amorphous material ($T_g \sim −20$ °C)[64,69]. Our results hence indicate that the selective incorporation of sulfur centres leads to tighter packing of polymer chains than in the all-oxygen case enabling semi-crystallinity and a regular polymer

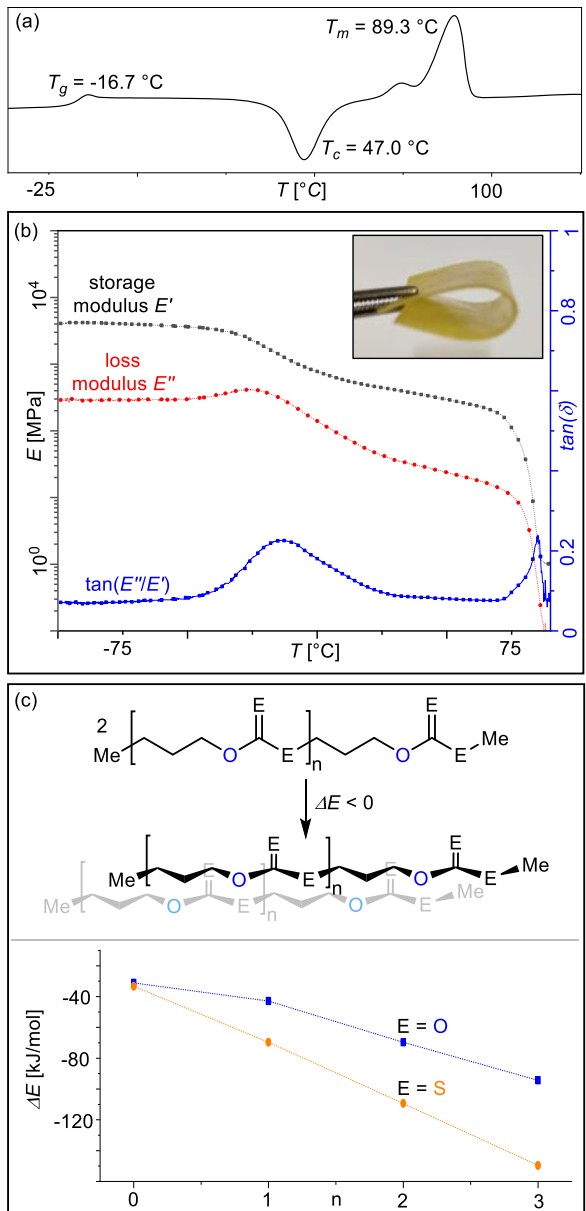

**Fig. 6 | Material properties. a** DSC curve of the 2nd heating cycle for polymer corresponding to Table 1, run #1. **b** Overlay plots of storage modulus *E'*, loss modulus *E''* and tan(*E''/E'*) for hot-pressed films (example photograph inlayed) with materials from Table 1 run #9 measured by DMA (tension film mode, 1 Hz, 1 °C per min). **c** Free energy release upon association of model oligomers assessed on the B3LYP/cc-pVDZ level of theory.

induced crystallinity[32,70]. AIM analysis reveals the absence of covalent interactions, indicating van-der-Waals forces to be the dominant form of attraction between chains. Attempted optimisation of dimers without dispersion correction hence results in dissociation of the chains. Consequently, more polarisable sulfur centres should result in enhanced van-der-Waals interaction compared to the oxygen analogues. Accordingly, the association energy for dimerisation decreases by ca. 25 kJ/mol on average per additional repeat unit n for E = O and by ca. 35 kJ/mol on average for E = S. Therefore, we suggest that increased interchain London dispersion contributes to sulfur induced semi-crystallinity in our systems. In comparison the COS/OX copolymer shows a significantly higher $T_m$ of 133 °C (versus 89.3 °C for the $CS_2$/OX copolymer) and enhanced hydrogen bonding interaction have been proposed to induce semi-crystallinity. Hence sulfur induced semi-crystallinity appears to arise through a combination of determinants exceeding the simple model outlined above.

## Discussion

In conclusion, we have demonstrated that moving from inter to intramolecularly cooperative catalysis unlocks the potential of $CS_2$ ROCOP to selectively yield sulfur-rich copolymers. Metal choice and proximity are responsible for limiting O/S scrambling side reactions that originate from metal-alkoxide chain-ends. The strategy also achieves the copolymerisation of previously unexplored monomer combinations yielding sulfur containing polymers from oxetanes. Moving from oxygen to sulfur renders our dithiocarbonates easier to depolymerise than the carbonate analogues as the produced heterocycle is less strained. Sequence selectivity is key to enable selective depolymerisation into dithiocarbonates and to obtain semi-crystalline materials to which increased interchain van-der-Waals interaction upon formal sulfuration contributes. Our report opens the door for the development of new main-group rich polymers with useful properties which were previously deemed inaccessible.

## Methods
### Materials

Solvents and reagents were obtained from commercial sources and used as received unless stated otherwise. Oxetane in particular was obtained from Sigma Aldrich as well as Fisher Scientific. Oxetanes, $CS_2$, PO and CHO were dried over calcium hydride at room temperature for 3 days followed by vacuum transfer (for CHO fractionally distilled under static vacuum) and three freeze pump thaw degassing cycles and stored inside an argon filled glovebox prior to use. Oxetane employed in Table 1 run #8, #9 and #10 was additionally dried over elemental sodium.

### Synthetic protocols

Synthesis of **A**: A solution of 2,3-dihydroxybenzaldehyde (5.72 g, 41.40 mmol) in dry DMSO (30 mL) was added dropwise over a period of 2 h to a suspension of NaH (2.19 g, 91.10 mmol) in dry DMSO (10 mL) under an $N_2$ athmosphere. Triethylene glycol ditosylate (9.50 g, 21.20 mmol) was added, and the resulting mixture was stirred for 48 h. Afterwards water (300 mL) was added, and the mixture was washed with $CHCl_3$ (100 mL). The organic layer was discarded, and the aqueous layer was acidified to pH 1 using 6 M HCl. The product was then extracted with chloroform (3 ×50 mL) and the combined organic layer was washed with HCl (1 M, 100 mL) and subsequently dried with $MgSO_4$. All volatile components were removed under vacuum, and the crude product was further purified using column chromatography (silica gel, chloroform:pentane:acetone, 65:40:10) yielding **A** as a pale-yellow solid (2.05 g, 5.25 mmol, 25% yield). The spectral data was in agreement with literature reports[73]. [1]H NMR (400 MHz, $CDCl_3$) δ 10.86 (s, 1H), 9.94 (s, 1H), 7.21 (dd, J = 7.8, 1.5 Hz, 1H), 7.17 (dd, J = 8.0, 1.6 Hz, 1H), 6.92 (t, J = 7.9 Hz, 1H), 4.22 (dd, J = 5.6, 4.2 Hz, 2H), 3.91 (dd, J = 5.6,

microstructure which is required to benefit from this effect. Although it has been previously reported that the formal substitution of sulfur for oxygen centres increases the crystallinity of polymers, the origins of this effect remain under debate[70–72]. Shedding light on the origin of crystallinity, we computationally assessed the intermolecular association between two molecules for a series of mono- to tetrameric $Me[(CH_2)_3\text{-}O\text{-}C(=E)\text{-}E\text{-}]_nMe$ model dithiocarbonates (E = S) in comparison to the all-oxygen carbonate (E = O) counterparts (Fig. 6, Supplementary Notes 10 in the Supplementary Information). We chose linearly stretched conformations to maximise the interaction between the respective chains. In all cases, free energy optimisation results in face on association of the oligomers which maximises the exposed surface area while minimising the formation of hydrogen bonds or S···S contacts which have previously been suggested to cause sulfur-

4.1 Hz, 2H), 3.77 (s, 2H). $^{13}$C NMR (101 MHz, CDCl$_3$) δ 196.04, 152.03, 147.34, 124.95, 121.17, 120.73, 119.46, 70.77, 69.55, 69.20. HRESI-MS positive mode: calculated [**A** + Na]$^+$ m/z calculated: 413.1212, found: 413.1335.

Synthesis of H$_2$**L'**: o-Vanillin (30.4 g, 0.2 mol) was dissolved in ethanol (500 mL). Cyclohexane-1,2-diamine (11.4 g, 0.1 mol) was added, and the mixture was stirred at reflux for 1 h. The mixtures was cooled to 0 °C causing the crystallisation of a yellow solid which was isolated by filtration and dried under vacuum to yielding H$_2$**L'** (10.5 g, 27.5 mmol, 28% yield). The spectral data was in agreement with lit-erature reports[74]. $^1$H NMR (400 MHz, CDCl$_3$) δ 8.33 (s, 1H), 6.94–6.67 (m, 3H), 3.80 (s, 3H), 3.36–1.25 (10H). $^{13}$C NMR (101 MHz, CDCl$_3$) δ 196.02, 152.01, 147.32, 124.93, 121.15, 120.71, 119.44, 70.75, 69.53, 69.18. $^{13}$C NMR (101 MHz, CDCl$_3$) δ 164.61, 151.37, 148.04, 122.96, 118.19, 117.72, 113.65, 72.16, 55.79, 32.80, 23.82. HRESI-MS positive mode: cal-culated [H$_2$**L'** + Na]$^+$ m/z calculated: 405.1790, found: 405.1961.

Synthesis of KOAc@18-crown-6: 18-crown-6 (264 mg, 1.00 mmol) and KOAc (196 mg, 2.00 mmol) were dissolved in THF (20 mL) and stirred for 1 h at room temperature. Afterwards all volatiles were removed in vacuum and the residual solid was extracted with DCM (20 mL). The resulting suspension was filtered and all volatiles were removed in vacuum yielding KOAc@18-crown-6 as a white solid (325 mg, 0.90 mmol, 90%). The spectral data was in agreement with literature reports[75]. $^1$H NMR (400 MHz, CDCl$_3$) δ 3.64 (s, 24H), 1.98 (s, 3H). $^{13}$C NMR (176 MHz, D$_2$O) δ 180.72, 69.79, 23.53. HRESI-MS positive mode: calculated [KOAc@18-crown-6 - OAc]$^+$ m/z calculated: 303.1215, found: 303.1247.

Synthesis of **LCrK**·H$_2$O: Inside a nitrogen-filled glovebox KOAc (55 mg, 0.56 μmol) was added to solution of the dialdehyde precursor **A** (219 mg, 0.56 μmol) in acetonitrile (10 mL) and the solution was stirred at room temperature for 30 min. Afterwards, 1,2-diaminocy-clohexane (64 mg, 0.56 μmol) and Cr(OAc)$_2$ (95 mg, 0.56 μmol) were added and the reaction mixture was stirred for further 16 h at room temperature. The complex was oxidised by exposure to air and by the addition of acetic acid (67 mg, 1.12 μmol) and the solution was stirred for 24 h open to air at room temperature. Afterwards, the suspension was filtered, and all volatiles were in vacuum. The resulting solid was washed with Et$_2$O (2 × 50 mL) and dried in vacuo to afford **LCrK**·H$_2$O as a brown powder (319 mg, 0.46 μmol, 82% yield). HRESI-MS positive mode: calculated [M - OAc]$^+$ 616.1279. Found 616.1249. Elemental Analysis: calculated C 51.94 %, H 5.52 %, N 4.04 %; found C 51.81 %, H 5.70 %, N 4.29 %. Following the analogous procedure **LCrNa**·H$_2$O was prepared with NaOAc (46 mg, 0.56 μmol) and obtained as a brown powder (159 mg, 0.26 μmol, 47% yield). HRESI-MS positive mode: cal-culated [M - OAc]$^+$ 600.1540. Found 600.1542; Elemental Analysis **LCrNa**·H$_2$O: calculated C 53.18%, H 5.65%, N 4.13%; found C 53.37%, H 5.44%, N 3.95%) and **LCrRb**·H$_2$O was prepared with RbOAc (81 mg) and obtained as a brown powder (224 mg, 0.30 μmol, 54% yield). HRESI-MS positive mode: calculated [M - OAc]$^+$ 662.0760. found 662.0806; El-emental Analysis **LCrRb**·H$_2$O: calculated C 48.69%, H 5.18%, N 3.79%; found C 48.50% H 5.33%, N 3.61%).

Synthesis of **L'Cr**: Under inert conditions H$_2$**L'** (1.0 g, 2.6 mmol) and Cr(OAc)$_2$ (0.44 g, 2.6 mmol) were dissolved in degassed acetoni-trile (200 mL) and stirred overnight at room temperature. Glacial acetic acid (0.30 mL, 5.22 mmol) was added and the solution was allowed to react for another 24 h open to air. Afterwards, the solvent was removed under reduced pressure and the crude product was washed three times with diethyl ether and dried under vacuum to obtain **L'Cr** as dark brown powder (1.03 g, 2.10 mmol, 80% yield). Ele-mental Analysis (**L'Cr**): calculated C 58.65%, H 5.54%, N 5.70%; found C 58.80%, H 5.60%, N 5.79%. HRESI-MS positive mode: [LCr-OAc]$^+$ m/z calculated: 432.1141, found: 432.1166.

**Synthesis of LZnK·2H$_2$O**: KOAc (55 mg, 0.56 μmol) was added to solution of **A** (219 mg, 0.56 μmol) in acetonitrile (10 mL) and the solution was stirred at room temperature for 30 mins. Afterwards,

1,2-diaminocyclohexane (64 mg, 0.56 μmol) and Zn(OAc)$_2$(H$_2$O)$_2$ (123 mg, 0.56 μmol) were added and the reaction mixture was stirred for a further 4 h at room temperature resulting in the precipitation of a yellow solid, which was isolated by filtration and dried in vacuo to afford **LZnK**·2H$_2$O as a yellow powder (110 mg, 0.16 μmol, 29%). $^1$H NMR (400 MHz, CDCl$_3$): δ[ppm] = 8.40–8,15 (s, 2H), 6.82 (d, J = 6.6 Hz, 2H), 6.74 (d, J = 7.1 Hz, 2H), 6.44 (t, J = 7.1 Hz, 2H), 4.39−3.46 (m, 15H), 2.07 (bs, 3H), 1.75−1.25 (s, 7H). No $^{13}$C NMR data was acquired due to the low solubility of the compound. Elemental Analysis: calculated C 50.49%, H 5.60%, N 4.21%; found C 50.81%, H 5.38%, N 4.29%. HRESI-MS positive mode: calculated [M - OAc]$^+$ 569.1032; found 569.0983.

Synthesis of OX$^{OR}$: General procedure: 3-methyl-3-oxetanemethanol (20.0 g, 0.20 mol) and the organic bromide (0.20 mol) were dissolved in benzene (50 mL). A 50 w% aqueous sodium hydroxide solution (40 g NaOH in 80 mL water) and tetra-n-butylammonium bromide (9.5 g, 0.03 mol) was added to the solution and stirred for 2 days. The organic layer was collected and washed with water (2 × 20 mL) and brine (1 × 20 mL) before being dried over mag-nesium sulfate. Removal of the solvent under vacuum yielded OX$^{OR}$ as colourless liquids. The spectral data were in agreement with literature reports.[62] OX$^{OBn}$: According to the general procedure 33.5 g of benzyl bromide were employed and OX$^{OBn}$ was obtained in 72% yield (144.0 mmol, 27.7 g). $^1$H NMR (400 MHz, CDCl$_3$): δ 4.60 (d, J = 5.8 Hz, 2H), 4.45 (d, J = 5.8 Hz, 2H), 3.60 (q, J = 7.0 Hz, 2H), 3.55 (s, 2H), 1.40 (s, 3H), 1.30 (t, J = 6.8 Hz, 3H). $^{13}$C NMR (151 MHz, CDCl$_3$) δ 138.30, 128.36, 127.59, 127.51, 80.06, 75.32, 73.28, 39.82, 21.37. HRESI-MS positive mode: calculated [M + Na]$^+$ 215.1048; found 215.1085. OX$^{OEt}$: According to the general procedure 21.8 g of ethyl bromide were employed and OX$^{OEt}$ was obtained in 65% yield (130.0 mmol, 16.92 g). $^1$H NMR (400 MHz, CDCl$_3$): δ 4.40 (d, J = 5.6 Hz, 2H), 4.26 (d, J = 5.6 Hz, 2H), 3.43 (q, J = 6.9 Hz, 2H), 3.38 (s, 2H), 1.21 (s, 3H), 1.11 (t, J = 7.0 Hz, 3H). $^{13}$C NMR (176 MHz, CDCl$_3$) δ 80.21, 75.87, 66.82, 39.86, 21.37, 15.04. HREI-MS positive mode: calculated [M]$^+$ 130.0994; found 130.1050. OX$^{OAll}$: According to the general procedure 24.2 g of allyl bromide were employed and OX$^{OAll}$ was obtained in 68% yield (136.0 mmol, 19.5 g). $^1$H NMR (400 MHz, CDCl$_3$): δ 5.82 (ddt, J = 16.2, 10.7, 5.5 Hz, 1H), 5.19 (d, J = 17.2 Hz, 1H), 5.10 (d, J = 10.4 Hz, 1H), 4.42 (d, J = 5.7 Hz, 2H), 4.26 (d, J = 5.7 Hz, 2H), 3.93 (d, J = 5.5 Hz, 2H), 3.39 (s, 2H), 1.22 (s, 3H). $^{13}$C NMR (101 MHz, CDCl$_3$) δ 134.71, 116.55, 79.73, 75.26, 72.12, 39.72, 21.25. HREI-MS positive mode: calculated [M]$^+$ 142.0994. Found 142.1082.

## Polymerisation protocol

Inside an argon filled glovebox, the catalyst, the monomers and an internal mesitylene standard were added to a flame dried vial equipped with a flame dried stirrer bar and sealed with a melamine cap containing a Teflon inlay. The vial was brought outside the glo-vebox and placed in a pre-heated aluminium block at the specified temperature for the specified time. At the specified end point of the reaction, the polymerisation mixture was cooled down to room temperature and extracted with CDCl$_3$ to determine the monomer conversion. The mixture was then added to 50 mL of MeOH and the polymer was isolated by centrifugation and dried in a vacuum oven set to 50 °C for 2 h. Alternatively the crude reaction mixture can also be cooled down to room temperature, causing the crystallisation of the polymer, which can then be isolated by centrifugation and air drying.

## Computational methods

Molecular periodic calculations were performed with the CRYSTAL17 program[76], using the B3LYP DFT functional in combination with Grimme type dispersion correction and employing the Gaussian-type atomic basis set cc-pVDZ (for more details see Supplementary Information)[77–80]. For the calculations of dimers of oligomer chains, the calculations were performed at the same level of theory but with the

Gaussian program; AIM analysis according to Bader was performed with the Multiwfn program[81,82].

## Data availability

The authors declare that the data supporting this study are available within the paper and the Supplementary Information file. All other data is available from the authors upon request.

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

## Acknowledgements

The VCI is acknowledged for a Liebig Fellowship for A.J.P. and a Ph.D. scholarship for C.G. Prof. Dr. Christian Müller and Prof. Dr. Rainer Haag are thanked for continuous support and valuable discussions. ZEDAT is thanked for computational resources, Biosupramol (FU Berlin), Thomas Rybak and Prof. Dr. Bernhard Schartel (Bundesanstalt für Materialforschung Fachbereich 7.5) as well as Prof. Dr. Andreas Thünemann (Fachbereich 6.5) are thanked analytical support.

## Author contributions

C.F.-W., B.R.M., M.R.S., C.G., P.P., A.J.P. jointly performed the experimental work and analysed the data. C.M. performed the DFT and evaluated the DFT calculations. A.J.P. conceived the project, wrote the paper and supplementary information and secured funding and directed the investigations.

## Funding

## Competing interests

The authors declare no competing interests.
