## [Peer Review File · Nature Communications]

Precise cooperative sulfur placement leads to semi-crystallinity and selective depolymerisability in CS₂/oxetane copolymersReviewers' Comments:

Reviewer #1:

Remarks to the Author:

The manuscript by Fornacon-Wood et al. reports on the copolymerization of CS₂ and oxetane using a Cr(III)-K based catalyst to prepare what appears to be a novel polydithiocarbonate. The work is well-designed and executed at a high level of competency, with A LOT of experimental detail provided on the conditions, catalyst performance and basic polymer properties. From a technical standpoint, I have no corrections, or concerns about the work presented in this manuscript. This catalyst system is very efficient to copolymerize CS₂ and oxetane. The only concern is on the overall impact of the work, since the resulting polydithiocarbonate has properties largely comparable to the similar polymers described in Figure 1 of the manuscript. The main impact lies in the significance of the catalyst system and the chemistry described in this submission. At this point, the evaluation is more subjective, as this referee would lean toward publication of this work in a more polymer focused journal, such as, *Macromolecules*.

Reviewer #2:

Remarks to the Author:

Review attached

The ring-opening copolymerizations of CS₂ and oxetanes in previously literatures are generally plagued by low linkage selectivity and small-molecule by-products. In this contribution, Plajer and co-workers reports the utilization of heterobimetallic Cr(III)-K, an intramolecularly cooperative catalytic system, can selectively deliver poly(dithiocarbonates) from CS₂ and oxetanes, yielding sulfur-rich copolymers with beneficial chemical and physical properties. Metal choice and proximity are demonstrated to be responsible for limiting O/S scrambling side reactions that originate from metal-alkoxide chain-ends. Overall, the results in this work are new and interesting, and the manuscript is generally well-written, so I recommend the publication in *Nature Communications* provided that the following points are addressed:

1. The monomer scope in current work is focused on oxetanes and cyclohexeneoxide. However, the copolymerization behavior of CS₂ with ethylene oxide and propylene oxide by heterobimetallic Cr(III)-K should also be studied, since both epoxides are more technologically/industrially important monomers.
2. The molecular weights of the resultant polymers are deviated from the theoretical ones. The authors attributed it to chain transfer events (i.e. with water in the catalyst itself and protic impurities in the monomers), cyclic polymer formation and other side reactions as often observed in related polymerization. Three questions here: (1) is it possible to remove the H₂O in the catalyst? (2) the protic impurities in the monomers should be easy to remove; (3) What does the other side reaction refer to? The elimination of negative effect of water and protic impurities might bring about controlled copolymerization.
3. Chain-end analysis is important for revealing the initiation and termination mechanisms. However, the determination of chain-end structure by MALDI-MS analysis is failed due to polymer decomposition. In this case, the authors should try ESI-MS analysis of low-molecule-weight polymer (M_n < 1000 g/mol).
4. The ratio of polymer to small-molecule by-products should be afforded in Tables.
5. The figure of tensile testing should be provided in supporting information.
6. Wrong citation: “The circumstance that we observe 99% polymer selectivity irrespective..... concerning the all-oxygen counterpart, i.e. CO₂/OX ROCOP.⁶¹”
7. “.....by Werner and co-workers for CS₂/epoxide ROCOP (Scheme S17)”: I cannot find Scheme S17 in supporting information.

8. Correct this sentence: "Testing LCrK in CS2 ROCOP of the ubiquitous epoxide Cyclohexeneoxide (CHO) also shows field leading performance (Table 3, run #5)."

Reviewer 1: The manuscript by Fornacon-Wood et al. reports on the copolymerization of CS₂ and oxetane using a Cr(III)-K based catalyst to prepare what appears to be a novel polydithiocarbonate. The work is well-designed and executed at a high level of competency, with A LOT of experimental detail provided on the conditions, catalyst performance and basic polymer properties. From a technical standpoint, I have no corrections, or concerns about the work presented in this manuscript. This catalyst system is very efficient to copolymerize CS₂ and oxetane.

We thank the reviewer for this positive feedback in favour of our manuscript.

The only concern is on the overall impact of the work, since the resulting polydithiocarbonate has properties largely comparable to the similar polymers described in Figure 1 of the manuscript. The main impact lies in the significance of the catalyst system and the chemistry described in this submission. At this point, the evaluation is more subjective, as this referee would lean toward publication of this work in a more polymer focused journal, such as, *Macromolecules*.

We agree with the reviewer that the material properties are comparable to the copolymer of carbonylsulfide and oxetane. Yet employing CS₂ (a liquid) rather than COS (a gas that is not commercially available in Europe) as a monomer is significantly easier to implement in a standard laboratory setup giving researchers that read our report a straightforward access to such polymers for further study which we believe is an important consideration in favour of the impact of our report.

Furthermore, as correctly identified by the reviewer the impact of our work stems primarily from the insights on (de)polymerisation catalysis. To further strengthen this aspect of our work and as requested by **Reviewer 2**, we have added additional data regarding the copolymerisation of propylene oxide (PO). Its copolymerisation is the focus of many research groups in the field as PO has particular technological relevance (see below).

Furthermore, another key message of our work is that selective copolymerisation enables selective depolymerisation of our polymers into cyclic dithiocarbonates therefore enabling polymer to monomer recycling and this has not been explored previously for this class of sulfurated oxetane copolymers. Scrambled CS₂/oxetane copolymers yield inseparable mixture of depolymerisation products preventing selective repolymerisation which is an increasingly important aspect of new polymerisation methodologies. Hence control over the sulfur atom placement enables new chemical utility. This may have been lost in our report. To highlight this aspect of our work we have changed the title of our report to:

Precise cooperative sulfur placement leads to semi-crystallinity and selective depolymerisability in CS₂/oxetane copolymers

Also, we have made the following change to the abstract of our work:

Furthermore, linkage selectivity is key to obtain semi-crystalline materials that can be moulded into self-standing objects as well as to enable chemoselective depolymerisation into cyclic dithiocarbonates which themselves can serve as monomers in ring-opening polymerisation.

On p5:

... into cyclic dithiocarbonates containing one O and two S atoms offering easy synthetic access to this class of heterocycles.

... forms a complex inseparable mixture of products. This implies that the sequence control achieved by LCrK is key to enabling chemoselective depolymerisation of the CS₂/OX copolymers.

Furthermore, we have added the following subsection headings to our report to highlight the individual aspects of our work:

CS₂/OX ROCOP by LCrK; Structure-selectivity and activity studies; Monomer scope of LCrK; Sulfuration and sequence-selectivity improve selective depolymerisation; Sulfuration and sequence-selectivity lead to semi-crystallinity

Reviewer 2:

The ring-opening copolymerizations of CS₂ and oxetanes in previously literatures are generally plagued by low linkage selectivity and small-molecule by-products. In this contribution, Plajer and co-workers reports the utilization of heterobimetallic Cr(III)-K, an intramolecularly cooperative catalytic system, can selectively deliver poly(dithiocarbonates) from CS₂ and oxetanes, yielding sulfur-rich copolymers with beneficial chemical and physical properties. Metal choice and proximity are demonstrated to be responsible for limiting O/S scrambling side reactions that originate from metal-alkoxide chain-ends. Overall, the results in this work are new and interesting, and the manuscript is generally well-written, so I recommend the publication in *Nature Communications* provided that the following points are addressed:

1. The monomer scope in current work is focused on oxetanes and cyclohexeneoxide. However, the copolymerization behavior of CS₂ with ethylene oxide and propylene oxide by heterobimetallic Cr(III)-K should also be studied, since both epoxides are more technologically/industrially important monomers.

The revised version of the manuscript now also features two runs exploring the ROCOP of CS₂ and propylene oxide (PO). The corresponding sections and table entries read as follows:

Figure 4: Monomer combinations investigated in CS₂ ROCOP with LCrK corresponding to **Table 3**.

Moving to the technologically relevant epoxide propylene oxide (PO) shows significantly more cyclic carbonate by-products (30% polymer selectivity) and scrambling (20% **OSS** links) at 80°C (**Table 3**, run #6) although decreasing the reaction temperature to 50°C leads to some improvements to 57% polymer and 54% **OSS** selectivity (**Table 3**, run #7). This **OSS** selectivity exceeds prior literature reports only reaching ≤16% dithiocarbonate links.^{31,35} Interestingly, 2D-NMR analysis reveals regioselective PO ring opening at the CH₂ position so that tertiary CHMe carbons sit adjacent to O-atoms in the **OSO** errors while secondary CH₂ carbons sit adjacent to S-atoms in the **SSS** errors confirming a scrambling mechanism as proposed by Werner and co-workers (Scheme S13).

Run	Loading ^a	t [h]	T [°]	TON ^b	Alt. [%] ^c	Polymer [%]	M _n [kDa] (Đ) ^d	T _{m/g/d,5%}
#6	1000 PO: 2000 CS ₂	2	80	1000	20	30	n.d.	n.d.
#7	1000 PO: 2000 CS ₂	16	50	480	54	57	6.56 (1.53)	T _g = 11.5 °C T _d = 147.3 °C

The corresponding data has been added to Section S8 of the revised version of the supporting information.

Figure S 70: Overlaid ¹H NMR spectra (400 MHz, CDCl₃, 25°C) of the crude reaction mixtures corresponding to table 3, run #6 and #7 showing less cyclic carbonate by-products going from 80°C (#6) to 50°C (#7).

Figure S 71: ¹H NMR spectrum (400 MHz, CDCl₃, 25°C) of the isolated polymer corresponding to table 3, run #7.

Figure S 72: $^{13}\text{C}\{^1\text{H}\}$ NMR spectrum (126 MHz, CDCl_3 , 25°C) of the isolated polymer corresponding to table 3, run #7.

Figure S 73: $^1\text{H}\text{-}^{13}\text{C}$ HMBC NMR spectrum (CDCl_3 , 25°C) of the isolated polymer corresponding to table 3, run #7.

Figure S 74: SEC trace corresponding to table 3, run #7.

Figure S 75: TGA data corresponding to table 3, run #7. $T_{d,5\%} = 147.3 \text{ °C}$.

Figure S 76: DSC data corresponding to table 3, run #7.

Furthermore, as pointed out in the main text the regioselectivity of the scrambled links shed further light on the O/S scrambling mechanism showing the regioselectivity of PO ring-opening at the CH₂ tail position is preserved throughout scrambling supporting a pathway alike *Werner's* mechanistic suggestion (Ref [35] of the main manuscript as well as Figure S13 of the revised version of the manuscript), which we believe adds additional valuable mechanistic information of CS₂ ROCOP in general.

Furthermore CS₂/EO ROCOP has been challenging from a safety perspective as ethylene oxide (EO) is a highly explosive and carcinogenic gas at room temperature. We installed a condensation line set-up and conducted EO ROCOP in high pressure resistant glass tubes instead of in melamine cap vials as for the other ROCOPs. EO sensors were put in place to ensure safety. Dosing of the EO was conducted by monitoring of the change in line pressure (with a known volume assuming ideal gas behaviour which we have found to be the safest set-up). EO was condensed into a pressure tube containing a catalyst CS₂ mixture in similar fashion as described in *Macromol. Rapid Commun.* 2021, 42, 2000472, the only report on CS₂/EO ROCOP in the literature. A picture of this set-up is shown below.

Reaction for 16h at 50°C, in analogy to the conditions employed for PO as explained above, resulted in approx. 10% EO conversion forming the product mixture shown below which can be assigned with help of the data presented in Macromol. Rapid Commun. 2021, 42, 2000472:

However, GPC analysis after precipitation from DCM/MeOH only shows oligomeric materials with $M_n < 2$ kDa.

During CS₂/EO ROCOP we also see a colour change from orange/brown to brown/green alongside the formation of a small amount of insoluble precipitate, an observation which has not been made in any other ROCOPs investigated in our report for which mixtures remain orange/brown and homogenous during catalysis. This observation as well as the low conversion indicates catalyst degradation during CS₂/EO ROCOP. Hence, we unfortunately find that our catalyst does not perform well for this monomer combination. To not deflect too much from the primary focus of our report we would prefer to summarize these findings on p6 of the revised version of our manuscript as follows:

.....Testing our catalyst in CS₂/ethylene oxide ROCOP under comparable conditions to PO unfortunately results in visible catalyst degradation.

2. The molecular weights of the resultant polymers are deviated from the theoretical ones. The authors attributed it to chain transfer events (i.e. with water in the catalyst itself and protic impurities in the monomers), cyclic polymer formation and other side reactions as often observed in related polymerization. Three questions here: (1) is it possible to remove the H₂O in the catalyst?

Unfortunately, prolonged drying under dynamic vacuum does not lead to the loss of the coordinated water. To clarify this circumstance we made the following addition to p2 of the revised version of the manuscript:

Attempts to remove H₂O via prolonged drying under dynamic vacuum at elevated temperatures were unsuccessful.

(2) the protic impurities in the monomers should be easy to remove.

We would like to thank the reviewer for making us reconsider this issue as the theoretical molecular weights in the table details of our initial submission have been wrong. Here we didn't take the circumstance appropriately into account that LCrK bears two rather than one initiating acetate and accordingly all theoretical weights have been corrected in the revised version of the manuscript. Furthermore, in table 1 where we now explicitly discuss the deviation of the obtained from the theoretical molecular weights in the main text the latter have now been added as a separate column in table 1 to the revised version of the manuscript.

Indeed, monomer purification is important for the outcome of CS₂/oxetane ROCOP. As stated in the supporting information of our initial submission CS₂ has been purified over CaH₂. Oxetane, the likely source of protic impurities due to its polar nature, has likewise been dried over CaH₂ for Table 1 runs #1 - #7 of our initial submission. In order to target higher molecular weights in run #8 and #9 of our initial submission, oxetane which was successively purified over CaH₂ and elemental Na was employed as stated in the supporting information of our initial submission. In order to further investigate the precise effect of the additional Na purifying step and hence the further removal of protic impurities we have added another run to Table 1 of the revised version of the manuscript, i.e. the equivalent of run #1 albeit with the doubly purified oxetane which indeed showed that when compared to singly purified

oxetane an approximate doubling in M_n is observed (Table 1 run #8 of the revised version of the manuscript and the corresponding GPC trace in Figure S14 of the revised version of the SI).

Run	Loading ^a	t [h]	T [°]	TON ^b	Alt. [%] ^c	Polymer [%]	M_n [kDa] (\mathcal{D}) ^d	$T_m/g_{d,5\%}$
#8	1:1000: 2000	8	1.5	1000	99	96	29.20 (1.83)	67.17

Other attempts to improve oxetane purity such as drying over molecular sieves and employing oxetane from another supplier did not result in any further improvements letting us hypothesise that side reactions other than chain-transfer with protic impurities (see answer to point (3) raised by the reviewer) limit the maximum achievable molecular weights.

Furthermore, we believe that the nature of the chain-ends is conceptually linked to the discussion of the deviations of obtained and theoretical molecular weights and investigations into the chain-end structures have been suggested later by the reviewer. Here we could identify acetate as well as primary alcohol chain ends (note that hypothetical xanthic acid chain ends are unstable towards CS_2 loss undergoing transformations into alcohols at ambient temperatures so formation of those can be excluded). To address all these facts, we have rewritten a substantial section of p3 of the revised version of our manuscript now reading:

..... thermodynamically favourable process. process. Attempts to obtain MALDI or ESI-MS data to investigate the polymer end-groups were unsuccessful. However, IR spectroscopy shows some ester end-groups ($\tilde{\nu}(\text{C}=\text{O}) = 1720.0 \text{ cm}^{-1}$) from acetate initiation to be part of the CS_2/OX copolymer which could be unambiguously identified by $^1\text{H}-^{13}\text{C}$ HMBC ($\delta(\text{C}^{\text{O}}=\text{O}, \text{Ester}) = 171.0 \text{ ppm}$ correlating to the OAc methyl group). Furthermore, the ^{31}P -NMR end-group test (SI Figure S17 and S18) allowed us to clearly identify primary alcohol chain ends and this suggests initiation via OX ring opening by the acetate coligands of LCrK and termination by protonation during work-up forming $\alpha\text{-OAc}, \omega\text{-OH}$ -functional chains. Yet the ^{31}P -NMR end-group test also indicates the formation of some $\alpha, \omega\text{-OH}$ -telechelic chains via initiation from H_2O or diol impurities. Accordingly, although molecular weights correlate to some extent with TON (i.e. lower TON in run #4 resulting in lower M_n) and M_n increases with reaction progress, theoretical (assuming initiation of both acetates) and observed molecular weights deviate significantly. We suspected that this is presumably due to chain transfer events with i.e. catalyst bound water or diol impurities in OX which is commonly observed in ROCOP.¹⁸ We hypothesized that the latter might occur, even though OX for run #1-#7 had been purified over CaH_2 . Hence, we employed OX which had been successively purified over CaH_2 and elemental Na (run #8) in a repeat to run #1. We indeed observed an approximate doubling of the obtained molecular weight ($M_n = 14.03 \text{ kDa}$ in run #1 to $M_n = 29.20 \text{ kDa}$ in run #8) which, although improved, is still lower than $M_{n,\text{theo}}$ (67.17 kDa) even when taking chain-transfer with the catalyst bound H_2O into account ($M_{n,\text{theo}} = 44.80 \text{ kDa}$) and this deviation becomes more pronounced for lower catalyst loadings. Other attempts such as drying over molecular sieves or testing OX from different suppliers to remove and avoid water or other protic impurities did not result in further improvement letting us hypothesise that side reactions other than chain transfer (such as cyclic polymer formation; *vide infra*) may also limit the molecular weights.^{18,46,58} LCrK also shows excellent performance (max. TON 7300) and activity at loadings as low as 0.003 mol% versus liquid monomers (run #10) producing polymers with a high maximum $M_n = 100.70 \text{ kDa}$ ($\mathcal{D} = 1.63$). State-of-the-art catalysis previously only achieved a maximum $M_n = 13.7 \text{ kDa}$ ($\mathcal{D} = 1.70$).³⁴

Structure-selectivity and activity studies

Next, we wondered whether the excellent performance of LCrK is ...

Also, the following information has been added to the supporting information section S1 and section S4 of the revised version of the supporting information:

Solvents and reagents were obtained from commercial sources and used as received unless stated otherwise. Oxetane in particular was obtained from Sigma Aldrich as well as Fisher Scientific. NMR spectra were recorded by using a Jeol JNM-ECA 400II, Bruker Advance 600 and 700 MHz spectrometer. ^1H and $^{13}\text{C}\{^1\text{H}\}$ chemical shifts are referenced to the residual proton resonance of the deuterated solvents.

Oxetanes, CS₂, PO and CHO were dried over calcium hydride at room temperature for 3 days followed by vacuum transfer (for CHO fractionally distilled under static vacuum) and three freeze pump thaw degassing cycles and stored inside an argon filled glovebox prior to use. Oxetane employed in table 1 run #8, #9 and #10 was additionally dried over elemental sodium.

As well as:

Figure S 15: IR spectrum of copolymer corresponding to table 1, run #4.

Figure S 16: Zoom into the ¹H-¹³C HMBC NMR spectrum (CDCl₃, 25°C) spectrum of copolymer corresponding to table 1, run #4.

End-group analysis by ³¹P NMR spectroscopy: A reported procedure by Marchessault (*Macromolecules* 1997, 30, 327-329) for the analysis of protic end-groups was followed. A polymer sample (20 mg of table 1 run #3) and a stock solution (40 μL, made of bisphenol A (400 mg), chromium(III)acetylacetonate (5.5 mg), and pyridine (10 mL)), in CDCl₃ (0.5 mL), were mixed in an NMR tube. Excess 2-chloro-4,4,5,5-tetramethyldioxaphospholane (40 μL) was then added to the NMR tube which was then shaken. The mixture was allowed to react for 30 min before analysis by ³¹P NMR.

Figure S 17: ^{31}P NMR (162 MHz, CDCl_3 , 25°C) end-group analysis of (top) CS_2/OX copolymer (table 1, run #3) and (middle, bottom) control experiments with propanol and propanthiol.

Figure S 18: Relative integrals in the ^{31}P (162 MHz, CDCl_3 , 25°C) NMR end-group analysis of the copolymer (table 1, run #3) corresponding to $2.10\ \mu\text{mol}$ protic end-groups in $20\ \text{mg}$ polymer. Note that (assuming $M_n = 15660\ \text{g/mol}$ as per GPC and $\alpha\text{-OAc}, \omega\text{-OH}$ -functional chains from acetate initiation followed by uniform propagation) $1.27\ \mu\text{mol}$ polymer-chains would have been employed, hence more OH chain ends are formed than exclusive formation of $\alpha\text{-OAc}, \omega\text{-OH}$ -functional chains would imply, which we infer to stem from the formation of some $\alpha, \omega\text{-OH}$ -bifunctional telechelic chains.

Furthermore it should be noted that the field of ROCOP is notoriously plagued with significant deviations of the obtained from theoretical molecular weights, even when the most impressive and field leading molecular weights are obtained (see for example <https://onlinelibrary.wiley.com/doi/abs/10.1002/anie.202104981> presenting unprecedented molecular weights which still deviate significantly from the theoretical ones) and the reasons behind this still remain under investigation

(3) What does the other side reaction refer to? The elimination of negative effect of water and protic impurities might bring about controlled copolymerization.

The effect of protic impurities has been discussed above in the response letter. “other side reactions” refers to the formation of macrocyclic polymers as attack of dithiocarbonate bonds by alkoxide chain ends can principally also yield macrocyclic polymers as depicted in the revised version of Figure S13. To experimentally support this hypothesis, we added GPC data to the revised version of our manuscript (Figure S27) after subjecting our polymers to intentionally formed alkoxide intermediates as described in Scheme S4 of our initial submission and here we observe a clear decrease in molecular weight. The corresponding sections in the revised version of the main manuscript now read:

On p3: ...other than chain transfer (such as cyclic polymer formation; *vide infra*) may also.....

Vide infra referring to on p4:..... Notably this also results in a drastic decrease of the molecular weights down to oligomers smaller than 1 kDa suggesting that alkoxide originated polymer attack decreases molecular weights. Importantly this observation could help explain why obtained molecular weights deviate from theoretical ones. As shown in Scheme S13 alkoxide addition and elimination to heterocarbonates of the same chain can in principle lead to cyclic polymer formation to decrease molecular weights in addition to chain-transfer events.....

And the following information has been added to the revised version of the supporting information:

Figure S 27: Overlaid GPC traces before and after scrambling experiment outlined above.

Scheme 1: (Top) O/S scrambling hypothesis by *Werner and Komber*. (Middle) Adapted hypothesis for CS₂/OX ROCOP explaining the formation of scrambled links and cyclic byproducts. (Bottom) Mechanistic suggestion for the formation of cyclic polymer.

3. Chain-end analysis is important for revealing the initiation and termination mechanisms. However, the determination of chain-end structure by MALDI-MS analysis is failed due to polymer decomposition. In this case, the authors should try ESI-MS analysis of low molecule-weight polymer ($M_n < 1000$ g/mol).

Unfortunately, ESI-MS analysis as suggested by the reviewer has also been unsuccessful which we have also attempted prior to our initial submission. We attribute this to the low polarity of our polymer. However, we have obtained information about the nature of the chain ends via the ³¹P NMR end group test as well as IR spectroscopy and 2D NMR shedding light on initiation and termination which we discussed above.

4. The ratio of polymer to small-molecule by-products should be afforded in Tables.

This has been added in the current version of the manuscript.

5. The figure of tensile testing should be provided in supporting information.

This data has been presented as Figure S12 of our original submission.

6. Wrong citation: “The circumstance that we observe 99% polymer selectivity irrespective..... concerning the all-oxygen counterpart, i.e. CO₂/OX ROCOP.61”

This has been corrected in the revised version of our manuscript to Ref. 63.

7. “.....by Werner and co-workers for CS₂/epoxide ROCOP (Scheme S17)”: I cannot find Scheme S17 in supporting information.

This has been corrected to:

... by Werner and co-workers for CS₂/epoxide ROCOP (Scheme S13).

Also Scheme S13 now not only shows the adapted mechanism but also Werner’s original suggestion as shown above in the response letter.

8. Correct this sentence: “Testing LCrK in CS₂ ROCOP of the ubiquitous epoxide Cyclohexeneoxide (CHO) also shows field leading performance (Table 3, run #5).”

This has been corrected to:

Testing LCrK in the ROCOP of CS₂ with Cyclohexenoxide (CHO) also reveals field leading performance (Table 3, run #5).

General: A few typographical errors have been corrected as part of the revision process which have been marked throughout the revised version of the manuscript and the supporting information. Additionally, the alkoxide intermediates in Figure 3 of our initial submission had one CH₂ unit too many which has also been corrected in the revised version of our manuscript.

Figure 3: (a) Zoom into the ¹H NMR (CDCl₃, 400 MHz) spectra of crude mixtures produced by the monocomponent LCrK catalyst (Table 1 run #1) and bicomponent variants (Table 2 run #4 and #7) alongside photographs of the produced polymers. (b) Structures of other catalysts. (c) Reaction pathways of propagation versus O/S scrambling for mono and bicomponent catalysts, X= OAc, [R_n] = polymer chain.

Furthermore, a data availability statement has been added to the revised version of the manuscript similar to a related article in *Nature communications* (<https://doi.org/10.1038/s41467-021-27377-3>).

Data availability

The authors declare that the data supporting this study are available within the paper and the Supplementary information File. All other data is available from the authors upon request.

Reviewers' Comments:

Reviewer #2:

Remarks to the Author:

The authors have addressed all the comments and issues raised by the Reviewers and carefully revised their manuscript. A significant number of valuable additional experiments have been conducted. Publish as is.